# Identification of High Personal PM$_{2.5}$ Exposure during Real Time Commuting in the Taipei Metropolitan Area



**Cheng-Yi Wang [1], Biing-Suan Lim [2], Ya-Hui Wang [3] and Yuh-Chin T. Huang [4,\*]**

[1] Department of Internal Medicine, Cardinal Tien Hospital and School of Medicine, Fu-Jen Catholic University, New Taipei City 23148, Taiwan; cywang@mospital.com

[2] Department of Education and Research, Cardinal Tien Hospital, New Taipei City 23148, Taiwan; bslim0316@gmail.com

[3] Medical Research Center, Cardinal Tien Hospital and School of Medicine, Fu-Jen Catholic University, New Taipei City 23148, Taiwan; yhwang531@gmail.com

[4] Department of Medicine, Duke University Medical Center, Durham, NC 27710, USA

\* Correspondence: huang002@mc.duke.edu

**Abstract:** There has been an increase in the network of mass rapid transit (MRT) and the number of automobiles over the past decades in the Taipei metropolitan area, Taiwan. The effects of these changes on PM$_{2.5}$ exposure for the residents using different modes of transportation are unclear. Volunteers measured PM$_{2.5}$ concentrations while commuting in different modes of transportation using a portable PM$_{2.5}$ detector. Exposure to PM$_{2.5}$ (median (range)) was higher when walking along the streets (40 (10–275) μg/m$^3$) compared to riding the buses (35 (13–65) μg/m$^3$) and the cars (15 (8–80) μg/m$^3$). PM$_{2.5}$ concentrations were higher in underground MRT stations (80 (30–210) μg/m$^3$) and inside MRT cars running in underground sections (80 (55–185) μg/m$^3$) than those in elevated MRT stations (33 (15–35) μg/m$^3$) and inside MRT cars running in elevated sections (28 (13–68) μg/m$^3$) ($p < 0.0001$). Riding motorcycle also was associated with high PM$_{2.5}$ exposure (75 (60–105 μg/m$^3$), $p < 0.0001$ vs. walking). High PM$_{2.5}$ concentrations were noted near the temples (588 ± 271 μg/m$^3$) and in the underground food court of a night market (405 ± 238 μg/m$^3$) where the eatery stalls stir-fried and grilled food ($p < 0.0001$ vs. walking). We conclude that residents in the Taipei metropolitan area may still be exposed to high PM$_{2.5}$ during some forms of commuting, including riding underground MRT.

**Keywords:** air pollution; particulate matter; subway; traffic-related pollution

## 1. Introduction

Epidemiologic studies have established an association between exposures to air pollution particles from mobile and stationary sources and human mortality and morbidity at concentrations of particles currently found in major metropolitan areas [1]. This association has been documented in numerous investigations around the world and is remarkably consistent [1–8]. The adverse effects of particulate matter (PM) include both pulmonary and extrapulmonary morbidity and mortality [9–11]. It is estimated that the daily cardiopulmonary mortality rose by 0.3% for each 10-μg/m$^3$ increase in PM$_{10}$ (particulate matter < 10 μm in aerodynamic diameter). For long term cardiopulmonary mortality, the estimate was 6% for each 10-μg/m$^3$ increase in annual average exposure to PM$_{2.5}$ (PM < 2.5 μm) [5]. The risk is especially high in the elderly and patients with chronic obstructive lung disease, asthma, coronary artery disease, congestive heart failure and cardiac arrhythmias [12–17]. The adverse pulmonary effects after PM exposure include greater hospital admissions, pulmonary infections, asthma attacks, and exacerbations of chronic obstructive pulmonary disease [12,18]. The extrapulmonary adverse effects of PM are primarily cardiac diseases [1,5,19,20] and vascular diseases (e.g., ischemic stroke) [21–24].

Most modern cities have significant air pollution issues related to particle emissions from road traffic and other anthropogenic sources. City residents are exposed to ground

level fine particulate matter ($PM_{2.5}$) from mobile sources during commuting. The ground level $PM_{2.5}$ concentrations are known to be higher than those reported by the fixed-site monitoring stations, which are 10–15 m above the ground [25–29]. Outdoor $PM_{2.5}$ concentrations below the height of 10-m buildings (three-story) were 10 to 20 times greater than those found at higher high-rise buildings, especially near busy roads [30]. Inhaled black carbon concentrations may be underestimated by the monitoring sites by as much as four to nine times [31]. Concentrations of black carbon while walking and riding a bicycle to work were up to six times those while riding a bus [32].

There have been studies on personal $PM_{2.5}$ exposure measured by portable light scattering or gravimetric detectors during commuting in different transportation modes in different cities, but the results were variable [27,29,32–38]. Multiple factors may explain the variabilities. The climate of the city can be a factor affecting how people choose their transportation mode. In tropical cities where the weather is more humid, rainy and hot year-round, people may be inclined to choose the transportation modes that involve the least outdoor exposure. In cities with moderate climate, residents may use more outdoor transportation modes, and thus potentially are exposed to more roadside air pollutants. The availability of air-conditioned buses and socioeconomic status of the population also affect people's preference. The traffic patterns, traffic volumes and driving conditions may also explain part of the variabilities. For example, driving a car was exposed to higher PM, but if the windows were closed with air conditioning on, the PM concentration inside the car decreased [27,34,37]. When biking routes shared the road with car lanes, especially when the traffic volumes were high, the bikers had higher PM exposure [32,34].

The Taipei metropolitan area in Taiwan has a subtropical climate. A study in 2008 showed higher personal $PM_{2.5}$ exposure for motorcycle commuters compared to riders of mass rapid transit (MRT), buses and cars [29]. Over the past decade, more MRT lines have been added and the ridership has been increasing. The average daily transport ridership in 2019 has reached over 2 million [39]. With trains running more frequently, PM produced from abrasion and wear of rail tracks, wheels and braking pads during the motion of the trains will increase [40–42]. In addition, the number of automobiles and motorcycles has continued to rise, despite the expansion of the MRT network, worsening PM produced from automobile traffic. These changes indicate that residents in the Taipei metropolitan area may continue to be exposed to higher $PM_{2.5}$ during commuting and an updated study to quantify the exposure is needed.

Although there have been studies on personal $PM_{2.5}$ exposure in cities that have different transportation infrastructures [32,43–47], these cities are located in different geographic regions with different climates. Their citizens have different lifestyles and cultures and tend to use modes of transport that are most convenient and economical. Therefore, it is essential to characterize the exposure to air pollutants in a specific urban environment so the health risk can be more accurately assessed.

In this study, we hypothesized that MRT riders in the Taipei metropolitan area were exposed to high $PM_{2.5}$ during commuting. We measured personal $PM_{2.5}$ exposure during commuting in different modes of transportation in the greater Taipei metropolitan area. The main goal was to provide an updated estimate of personal $PM_{2.5}$ exposure. The results can help identify high risk subpopulations that can be the focus of future studies on $PM_{2.5}$-associated health effects.

## 2. Materials and Methods

The greater Taipei metropolitan area includes Taipei City and New Taipei City (Figure 1). The Metro routes have six lines: Wenhu line (brown), Tamsui-Xinyi line (red), Songshan-Xindian line (green), Zhonghe-Xinlu line (orange), Bannan line (blue), and Circular line Phase I (yellow). The size of the area is 2325 km$^2$ (Taipei City: 272 km$^2$ and New Taipei City: 2053 km$^2$). As of 31 March 2019, the populations of Taipei City and New Taipei City were 2,663,425 and 3,998,883, respectively, or slightly more than one quarter of the population of Taiwan [48].

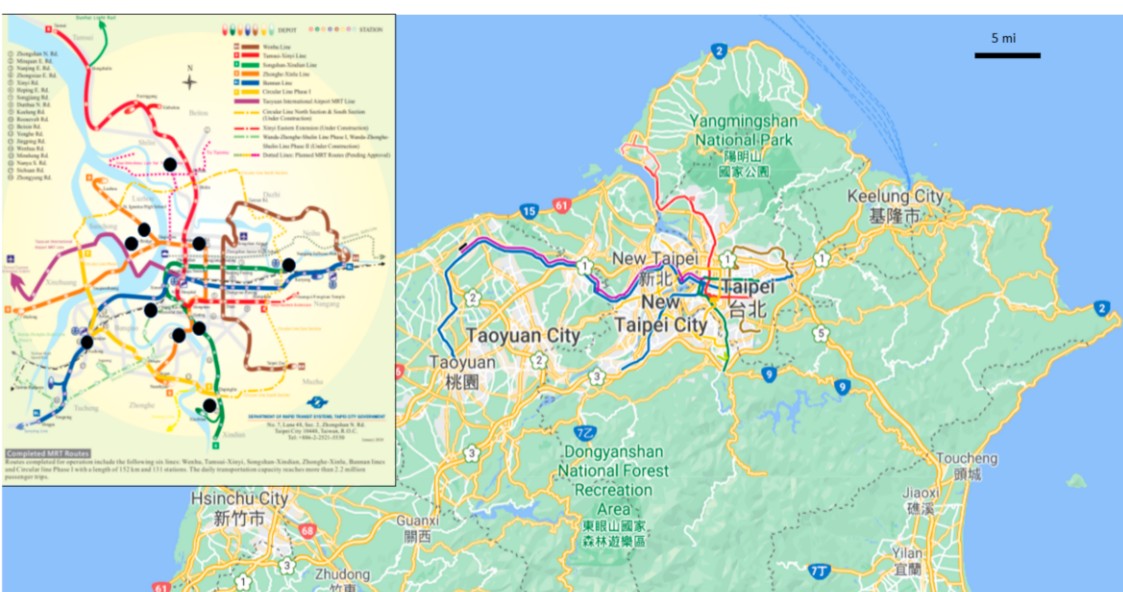

**Figure 1.** A map of northern Taiwan showing the location of Taipei City and New Taipei City. The inset is an enlarged map of the Metro lines. The black dots are the approximate locations of the EPA monitoring stations.

$PM_{2.5}$ concentrations to which a commuter was exposed were measured using a portable $PM_{2.5}$ detector (Temtop P600 Air Quality Laser Particle Detector, Elitech Technology, Inc., Milpitas, CA, USA). The detector is equipped with a laser particle sensor, and its operating environments include a temperature range: 0–50 °C; relative humidity range: 0–90%; atmospheric pressure: 1 atm; $PM_{2.5}$ measurement range: 0–999 $\mu g/m^3$ with a resolution of 0.1 $\mu g/m^3$. The time resolution is 1 min. The laser sensor used in this detector was that same as the one used in another detector (Temtop LKC-1000S+). The sensor was evaluated in the laboratory and in the field with the Federal Equivalent Method (FEM)-Grimm as the standard [49,50]. Taking the average of the three linear equations from the field test for the three detectors like ours, one may derive the following linear equation to make correction: [Adjusted $PM_{2.5}$] = 0.678 × [measured $PM_{2.5}$] + 3.298 (average $R^2$ = 0.915) [49]. Based on this equation, the measured $PM_{2.5}$ concentrations will need to be adjusted downward by about 30% [49]. The precision was very good with low intra-modal variability (~7%). The climate condition had minimal effect on the sensors' precision up to a relative humidity of 65%. We recorded 3 readings per measurement and took the average. The detector was not calibrated against FEM-GRIMM or Tapered Element Oscillating MicroBalance (TEOM) since they were not available to us. Instead, we recorded $PM_{2.5}$ concentrations in a small air-conditioned room before each trip to ensure that the readings were consistent. The field test showed that this detector had low intra-model variability (~7%) [49]. The $PM_{2.5}$ concentrations in the room were 5–10 $\mu g/m^3$. We used the basic Quality Assurance/Quality Control (QA/QC) method to validate the collected data. Negative values and invalid data-points were eliminated from the dataset. Five volunteers were recruited to take the trips and perform the $PM_{2.5}$ measurements. We used a small number of volunteers to decrease the interobserver variability. The same detector was used for the entire study to minimize the variability among different units, [49,50]. We measured $PM_{2.5}$ concentrations at only low wind condition.

The study was conducted between April and July 2019. The months from April to July usually have the lowest ambient $PM_{2.5}$ in northern Taiwan, as ambient air quality during these months tends to better and is least influenced by cross-border pollution from China [51]. Volunteers were recruited and asked to hold the detector at the midchest level parallel to the ground with the sensor facing away from the body during the measurements.

Ambient $PM_{2.5}$ data were obtained from the nearest monitoring stations set up by Taiwan Environmental Protection Agency [52]. The sampling ports of these stations are

located at a height of 15–20 m above the ground. Ambient $PM_{2.5}$ concentrations reported by the monitoring stations are measured using the β-ray attenuation method and Tapered Element Oscillating Microbalance Technology (TEOM). The particulate analysis instrument automatically measures the concentrations of $PM_{10}$ and $PM_{2.5}$ with a screening device and a mass calculation system [53].

### 2.1. MRT

The MRT system in the Taipei metropolitan area consists of five lines covering 131.1 km at the time of the study. The volunteers took trips crossing the Taipei metropolitan area. The trips lasted between 30–90 min and were taken during the morning (7:00 a.m.–9:00 a.m.) and afternoon (5:00 p.m.–7:00 p.m.) rush hours as well as off-peak times (10:00 a.m.–4:00 p.m.).

### 2.2. Walking

The volunteers measured $PM_{2.5}$ while walking along the streets and inside a night market. This night market in the Taipei region consists of a street level section and an underground quarter with individual stalls that sell clothing, drinks and a great range of snacks and food that are grilled or fried on site.

### 2.3. Bus

There is an extensive bus system in the Taipei metropolitan area. The volunteers rode buses to and from work or as connectors to the MRT stations. All buses in the metropolitan Taipei area have air conditioning and the air conditioning was on when the study was conducted (between April and July).

### 2.4. Motorcycle

The volunteers rode a motorcycle to and from work and to other destinations. For safety reasons, $PM_{2.5}$ concentrations were measured only when the motorcycle was idle, for example, waiting for red lights.

### 2.5. Private Car

The volunteers drove cars to and from work and to other destinations. The volunteers were asked to turn on the air conditioning and set the ventilation in the re-circulation mode with the windows closed while driving.

### 2.6. Recording of Environmental Conditions

Each volunteer also recorded the time of the day, details of the routes including street names, names of the MRT stations and any environmental characteristics that may affect $PM_{2.5}$ concentrations, including temples, roadworks, crowdedness and food vendors. The weather conditions at the time of the measurements including humidity, temperature and rain precipitation from the Central Weather Bureau of Taiwan were also recorded.

### 2.7. Statistical Analysis

All data were expressed as median and range or mean and standard deviation (SD). Comparisons among different conditions were performed using non-parametric multiple comparisons Wilcoxon rank sum test (JMP 13, SAS Inc., Cary, NC, USA). $p < 0.05$ was statistically significant.

## 3. Results

Ambient $PM_{2.5}$ concentrations reported by nearby monitoring stations during the study period ranged from 2–38 $\mu g/m^3$ with a median of 14 $\mu g/m^3$. Monthly average $PM_{2.5}$ concentrations for April, May, June, and July of 2019 from 10 monitoring stations in the Taipei metropolitan area were 19, 14, 12 and 12 $\mu g/m^3$, respectively.

Street level $PM_{2.5}$ concentrations near the monitoring stations were measured by the portable detector. Ambient $PM_{2.5}$ concentrations reported by the monitoring stations were recorded simultaneously. A total of 24 pairs of measurements were performed on different days at five different stations. The scattered graph is shown in Figure 2. The dotted line represents linear regression. The linear equation is S = 0.958 M + 23.353, where S is street level $PM_{2.5}$ concentrations and M is monitoring station $PM_{2.5}$ concentrations ($R^2$ = 0.275, $p$ = 0.006). The poor association was likely due to the difference in the measurement of $PM_{2.5}$ reported by the monitoring station at a height of 10–15 m and those measured at the ground level. $PM_{2.5}$ concentrations near the ground could be 10–20 times higher than those found 10 m above the ground [30]. The discrepancy could also be due to the different methods used to measure $PM_{2.5}$. The personal $PM_{2.5}$ levels were measured by the light scattering method while the ambient $PM_{2.5}$ levels were measured at each monitoring station by the TOEM method by Taiwan EPA.

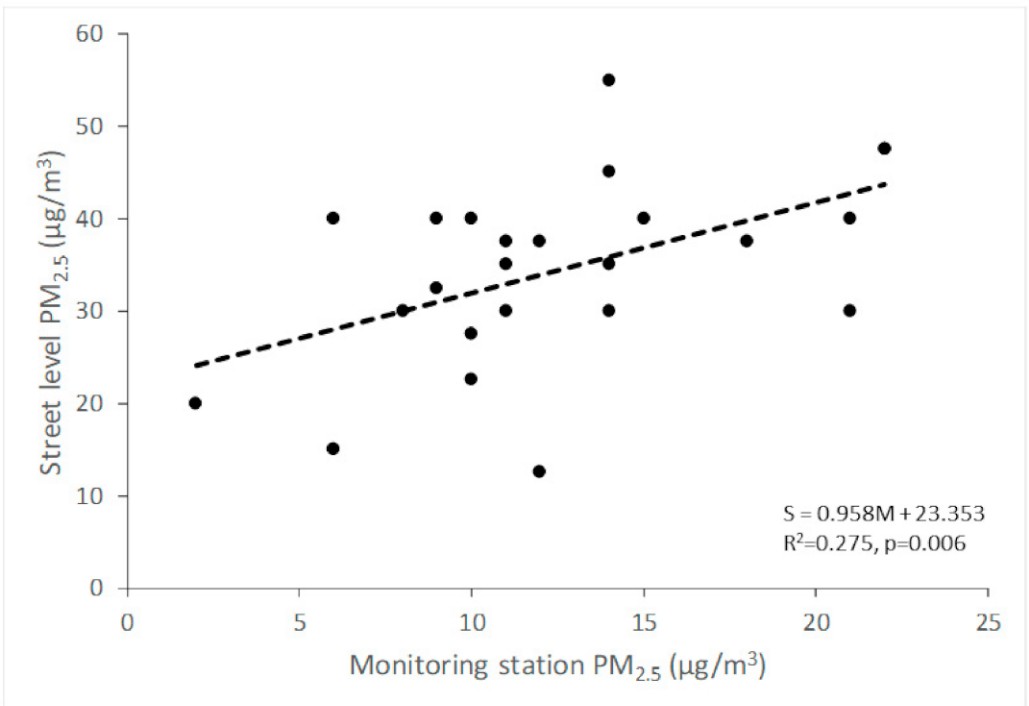

**Figure 2.** Correlation between $PM_{2.5}$ concentrations at the street level measured by the hand-held detector and those measured by the nearest monitoring stations.

The median $PM_{2.5}$ concentration when walking on the sidewalks of the city streets was 40 $\mu g/m^3$ (range 10–275 $\mu g/m^3$, $n$ = 216 measurements). Table 1 shows an example of a short walk from home to the workplace (hospital) on a sunny day with a reported temperature of 26 °C and relative humidity of 69%. Ambient $PM_{2.5}$ was 21 $\mu g/m^3$. Note the concentration rose at the intersections of major roads. The concentration increased at the intersection in part because of the increased automobile traffic that generated more exhaust and road dust. Table 2 shows an example of a longer walk from the workplace to home on a rainy day with a reported temperature of 24 °C and relative humidity of 77%. Ambient $PM_{2.5}$ was 5 $\mu g/m^3$. Note that $PM_{2.5}$ concentrations spiked near a temple. Table 3 shows a walking trip through the Shilin night market. The shaded area indicates locations inside the underground food court. Figure 3 shows the route map. The capital letters represent measurement points. The ambient $PM_{2.5}$ concentration was 5 $\mu g/m^3$. The Shilin night market is one of the largest night markets in Taiwan. It has many vendors and shops for durable goods and clothing as well as a famous food section with many eatery stalls. $PM_{2.5}$ concentrations in general were higher near food vendors and stalls (15:20 and

15:25 time points). The concentrations of $PM_{2.5}$ were especially high near a temple and in the underground food court where there are food stalls serving stir-fried and grilled food. Average $PM_{2.5}$ concentrations near the temple and inside the underground food court were $588 \pm 271$ μg/m$^3$ and $405 \pm 238$ μg/m$^3$ respectively.

**Table 1.** An example of a short walk from home to work (hospital) on 10 April 2019.

| Time of the Day (hr:min) | Location | PM$_{2.5}$ (μg/m$^3$) |
|---|---|---|
| 8:07 | In the alley | 65 |
| 8:10 | Intersection of major roads | 95 |
| 8:15 | Intersection of major roads | 100 |
| 8:17 | Hospital 1st floor | 65 |
| 8:20 | Office in the hospital | 57.5 |

**Table 2.** An example of a longer walk from the hospital to home on 10 April 2019.

| Time of the Day (hr:min) | Location | PM$_{2.5}$ (μg/m$^3$) |
|---|---|---|
| 14:21 | In front of the hospital | 24 |
| 14:25 | Bus stop by a major road | 36 |
| 14:36 | Bus stop by a major road | 22.5 |
| 14:40 | Crossing the road | 45 |
| 14:42 | Walking by a temple | 85 |
| 14:46 | At family courtyard | 42.5 |
| 14:48 | Inside the house | 20 |

**Table 3.** PM$_{2.5}$ concentrations during a walk across a business district.

| Time of the Day (hr:min) | Location | PM$_{2.5}$ (μg/m$^3$) |
|---|---|---|
| 16:32 | A (Fude Rd/Daxi Rd intersection) | 30 |
| 16:35 | B (Daxi Rd/Dabei Rd intersection) | 40 |
| 16:38 | C (Danan Rd) | 35 |
| 16:57 | D (Outside a Mazu temple) | 700 |
| 17:05 | E Night market entrance | 40 |
| 17:10 | F (Fried oyster cake stand) | 400 |
| 17:15 | F Fried chicken patty stand | 150 |
| 17:17 | F Grilled steak stand | 95 |
| 17:20 | G (Road by the night market) | 85 |
| 17:25 | H (night market/Jihe Rd Intersection) | 80 |
| 17:27 | I (Jihe Rd/Chengde Rd intersection) | 50 |
| 17:30 | J (Jihe Rd/Xiaoshi St intersection) | 30 |

A total of 34 MRT trips were taken. In the underground MRT sections, the median concentrations of $PM_{2.5}$ in the stations and in the metro cars were 75 μg/m$^3$ (range: 30–210 μg/m$^3$, $n$ = 92 measurements) and 80 μg/m$^3$ (range: 53–185 μg/m$^3$, $n$ = 188 measurements), respectively. Both were higher than the $PM_{2.5}$ concentration when walking along the streets ($p < 0.0001$). Figure 4 shows $PM_{2.5}$ concentrations during a trip that used two different underground MRT lines. Red dots are $PM_{2.5}$ concentrations reported by the nearest monitoring stations: Xindian (2 μg/m$^3$) and Guting (6 μg/m$^3$).

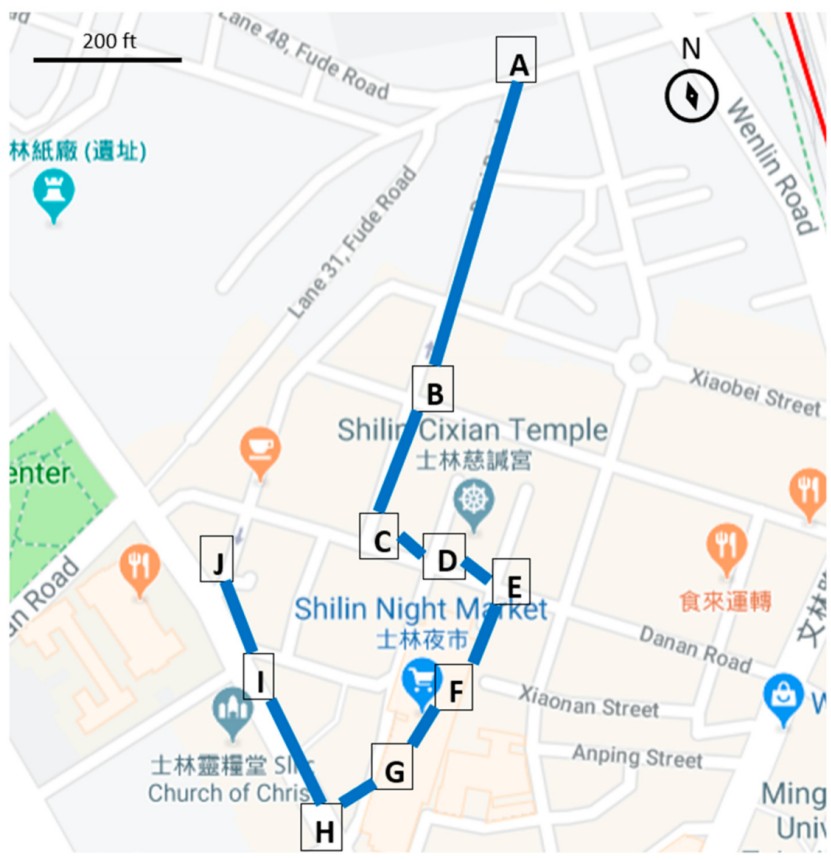

**Figure 3.** PM$_{2.5}$ concentrations during a walk across a business district that includes a night market and a temple on 27 April 2019.

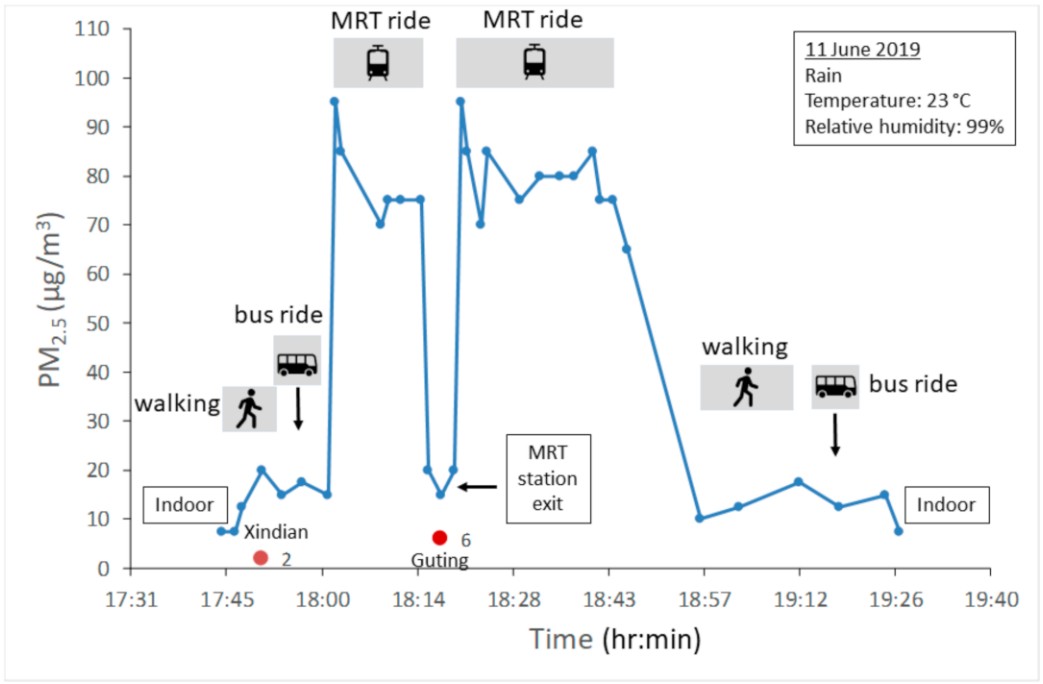

**Figure 4.** PM$_{2.5}$ concentrations during a trip from home to the workplace that used primarily two underground mass rapid transit (MRT) lines. MRT: mass rapid transit.

When the trains were traveling in the elevated MRT sections that were about 10–15 m above the ground, the concentrations of $PM_{2.5}$ in the stations were lower than those in underground stations (median: 33 μg/m³; range: 15–35 μg/m³, $n$ = 10 measurements, $p < 0.0001$ vs. $PM_{2.5}$ in underground MRT stations) and in trains running in the elevated sections (median: 20 μg/m³; range: 13–68 μg/m³, $n$ = 52 measurements, $p < 0.0001$ vs. $PM_{2.5}$ in underground MRT trains). Both were also lower than the $PM_{2.5}$ concentrations when walking along the streets ($p$ = 0.0025 for elevated MRT stations and $p < 0.0001$ for elevated MRT cars). Figure 5 shows $PM_{2.5}$ concentrations during a trip on an MRT line with both underground and elevated sections. Red dots are $PM_{2.5}$ concentrations reported by the nearest monitoring stations—Songshan (16 μg/m³), Zhongshan (17 μg/m³), Shilin (10 μg/m³) and Tamsui (9 μg/m³). It was evident that when the train emerged from the underground section, the $PM_{2.5}$ concentrations inside the train decreased.

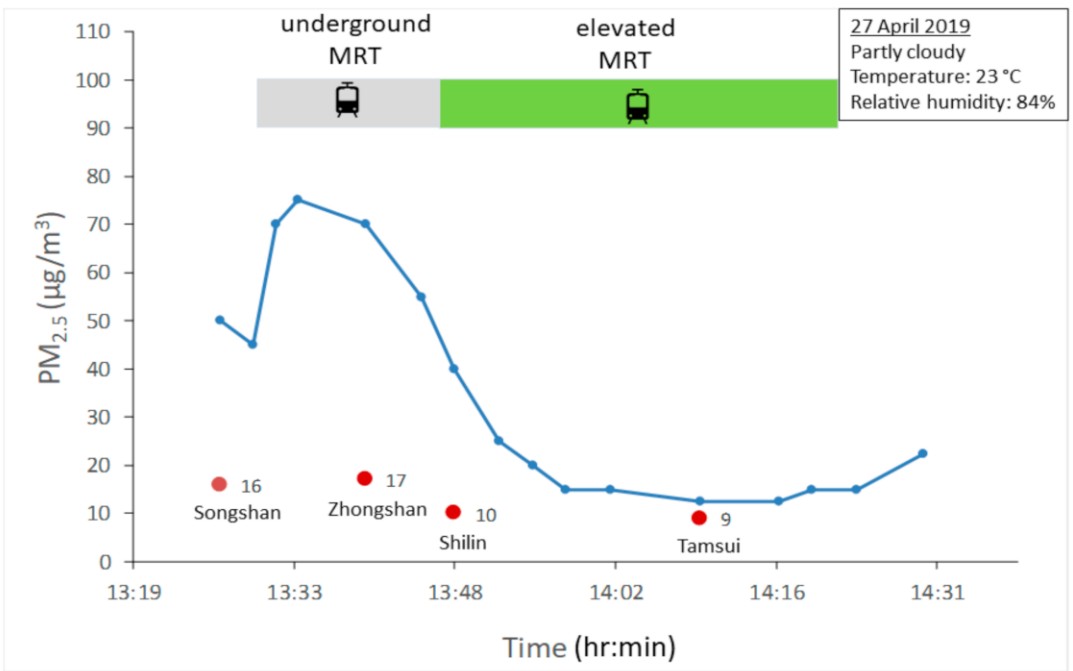

**Figure 5.** $PM_{2.5}$ concentrations during a trip that used an MRT line with underground and elevated sections.

The median $PM_{2.5}$ concentration inside the buses was 35 μg/m³ (range: 13–65 μg/m³, $n$ = 38 measurements), lower than that of walking ($p$ = 0.0025). At the time of the measurements, all buses were running air conditioning. Figure 6 shows a representative trip by bus. Red dots are $PM_{2.5}$ concentrations reported by the nearest monitoring stations: Xindian (4 μg/m³), Yonghe (7 μg/m³), Banqiao (7 μg/m³), Wanhua (6), Cailiao (10 μg/m³) and Sanchong (5 μg/m³). At the bus stops (on the roadside), the $PM_{2.5}$ concentrations were higher than those inside the bus.

A total of three motorcycle trips were taken. The median $PM_{2.5}$ concentration was 75 μg/m³ (range: 60–105 μg/m³, $n$ = 21 measurements, $p$ = 0.0011 vs. walking). Figure 7 shows a typical trip. Red dots are $PM_{2.5}$ concentrations reported by the nearest monitoring stations—Datong (20 μg/m³) and Songshan (17 μg/m³). The $PM_{2.5}$ concentration increased quickly by several fold after the ride began. Throughout the trip, the $PM_{2.5}$ concentrations remained high. In Taiwan, motorcyclists frequently ride on the car lanes. Therefore, they would be exposed to exhaust from the cars and motorcycles more directly than pedestrians and bus riders.

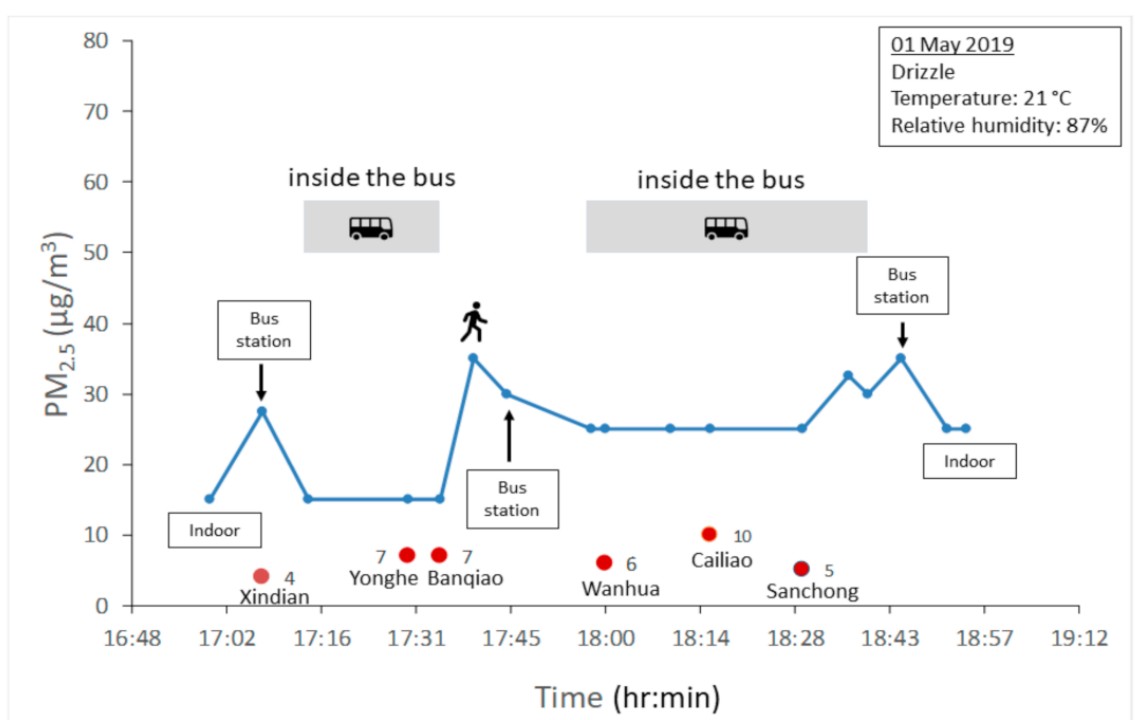

**Figure 6.** PM$_{2.5}$ concentrations during a bus ride.

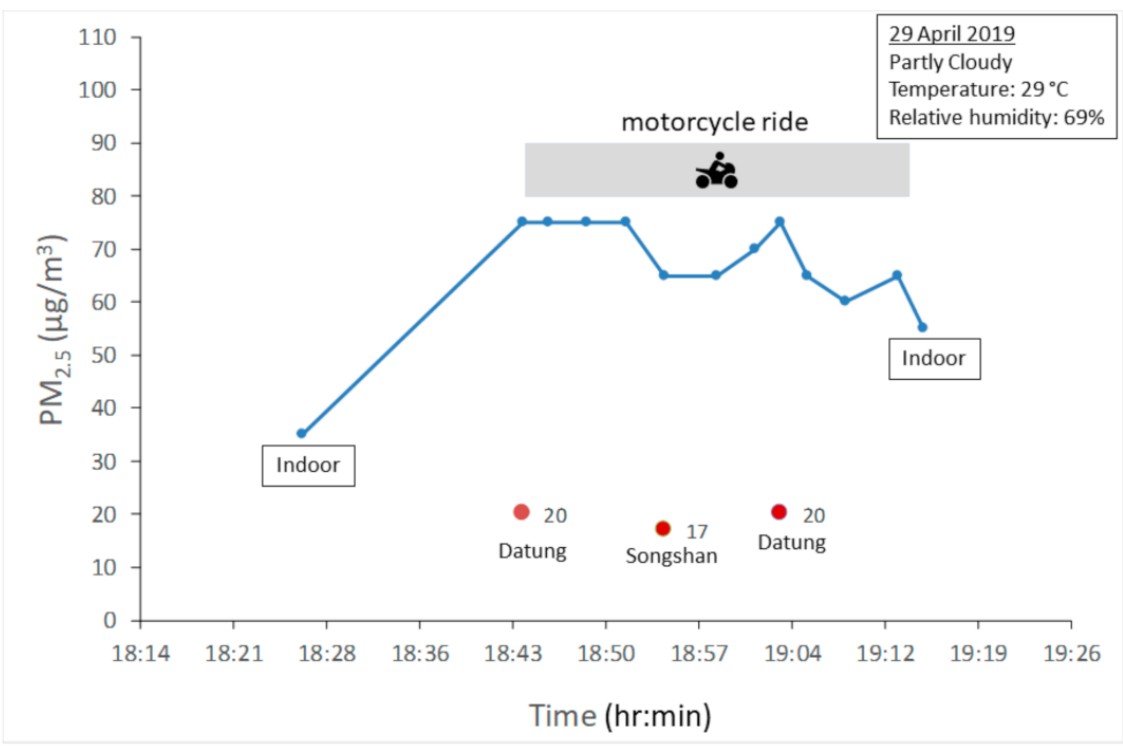

**Figure 7.** PM$_{2.5}$ concentrations during a motorcycle ride on a major road.

There were five car trips. The PM$_{2.5}$ concentrations were lower compared to those during walking (median: 15 µg/m$^3$; range: 8–80 µg/m$^3$, *n* = 111 measurements, *p* < 0.0001 vs. walking).

Figure 8 shows the comparison of personal PM$_{2.5}$ concentrations among different modes of transport. MRT station (underground), MRT car (underground) and motorcycle

were higher than walking along roadside (*). MRT station (underground) and MRT car (underground) are higher than MRT station (elevated) and MRT car (elevated) (#). Private cars and buses are lower than walking along the roadside (§).

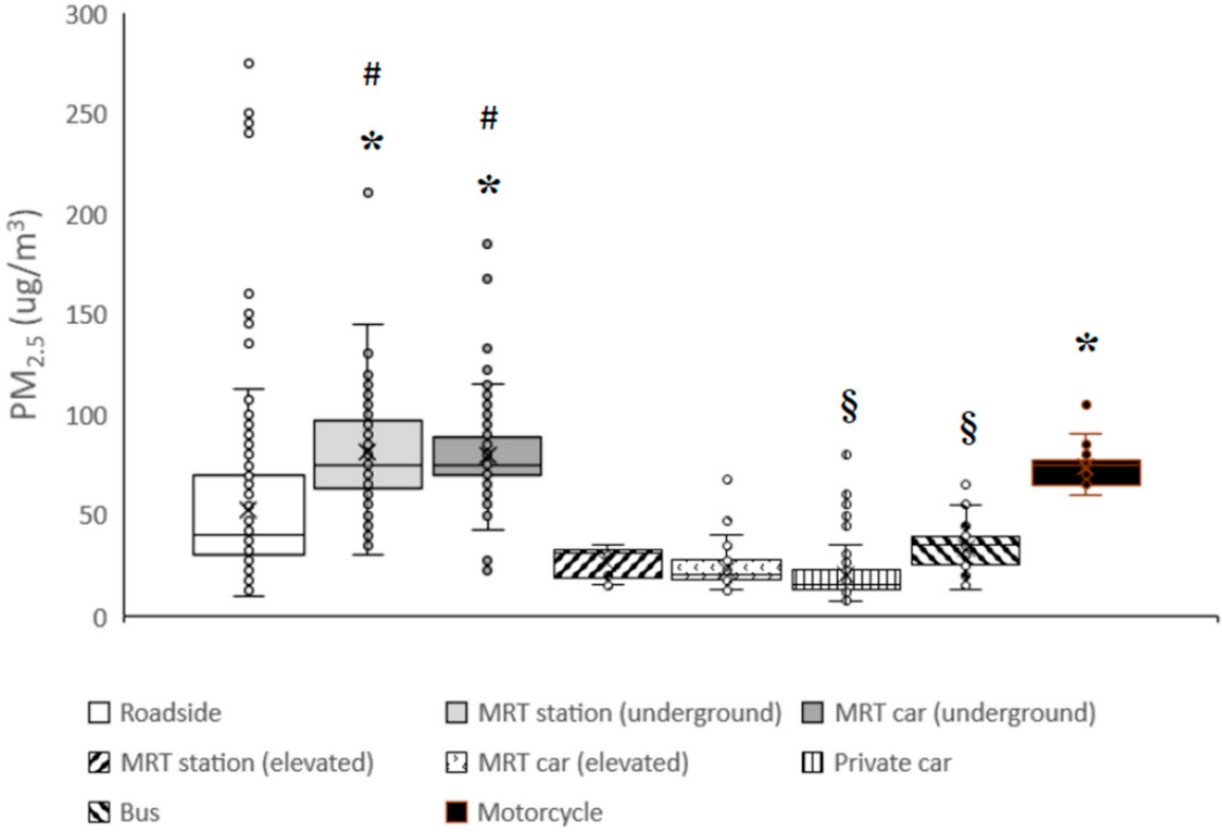

**Figure 8.** Comparison of $PM_{2.5}$ concentrations among different modes of transportation.

## 4. Discussion

Our study showed high $PM_{2.5}$ concentrations inside the underground MRT platforms and the trains (75 $\mu g/m^3$ and 80 $\mu g/m^3$). With the 30% downward adjustment for the $PM_{2.5}$ concentrations measured by our portable detector, these results were slightly lower than those from a previous study that showed an average $PM_{2.5}$ concentrations of 75.4 $\mu g/m^3$ in the winter and 56.2 $\mu g/m^3$ in the summer on the platforms of 10 most populous underground MRT stations in Taipei [54]. Assuming an MRT rider has a minute ventilation of 5 L/min and spends an average of 1 h/day in MRT, the rider may inhale more than 8 mg of $PM_{2.5}$ a year from MRT alone [55]. A resting minute ventilation is used for the calculation because when the $PM_{2.5}$ concentrations were measured, the MRT riders were stationary. The minute ventilation can increase when the riders walk around increasing the inhalation dose. MRT is built for convenient and efficient travel and has a goal of reducing automobile traffic and air pollutants produced by mobile sources. Although electric-powered, MRT generates its own air pollutants due to abrasion and wear of rail tracks, wheels and braking pads caused during the motion of the trains. The concentrations of PM in the MRT system were found to be higher than those measured by ambient monitors [43–45]. High levels of $PM_{2.5}$ had been shown in European and South Korean cities, especially in the underground subway systems [41,44–46,56]. Such PM contains abundant elemental iron, total carbon, crustal matter, secondary inorganic compounds, insoluble sulphate, halite and trace elements [43,44,46,56] and has similar toxic effects to health compared to PM from other mobile or fixed sources [46].

The high $PM_{2.5}$ concentration in MRT has two potential implications. First, it may discourage the residents to use the MRT in favor of other modes of transportation associated with lower $PM_{2.5}$, such as cars. In Taipei metropolitan area, the number of automobiles and motorcycles has increased over the past decade despite the expansion of the MRT network. While there are multiple reasons for this phenomenon, improving air quality in the MRT would be a good incentive for car drivers and motorcyclists to switch to riding MRT. This should be a major focus for city managers. Second, the MRT riders who spend a long time every day commuting to work can have significant cumulative exposure to high $PM_{2.5}$. This would increase their risk for chronic cardiopulmonary diseases associated with $PM_{2.5}$, such as coronary artery disease, chronic obstructive pulmonary disease (COPD), asthma and lung cancer (in particular adenocarcinoma) [5,9,57].

Walking is a basic mode of transportation and is promoted in many cities. In some European cities, walking has been associated with lower $PM_{2.5}$ exposure compared to taking cars or buses [33,35]. However, in Salt Lake City, Utah, USA, pedestrians received a higher $PM_{2.5}$ dose and had higher rates of exposure than commuters using automobiles or public transportation [34]. Our study showed walking along the streets was exposed to higher $PM_{2.5}$ concentrations than riding air-conditioned cars and buses. The differences among these studies may be due to traffic density and the use of air conditioning in the automobiles and public transportation. The nearby environment for the pedestrians is also important. For example, if one walks by temples, the $PM_{2.5}$ concentration can increase many folds due to incent burning. Our study also showed when one was inside the food court of a night market where stir frying and grilling are used to prepare food, exposure to $PM_{2.5}$ can be quite significant due to inadequate ventilation. The health effects from exposure to high $PM_{2.5}$ generated from the temples are well documented [58–61]. The cardiopulmonary health impact from exposure to high $PM_{2.5}$ concentration in cooks and waiter/waitresses who work in the eatery stalls deserves further study.

Motorcycle is a common mode of transportation in many tropical and subtropical Asian countries, including Taiwan. Motorcycle traffic contributes to roadside $PM_{2.5}$ [28,29,62]. At the same time, motorcyclists are exposed to exhaust from other motor vehicles as well as their own motorcycles, especially during busy traffic hours [27,29,63]. In our study, there were three trips taken with 21 measurements performed at different stops during the trips. All routes were on the thoroughfares in the Taipei metropolitan area. The $PM_{2.5}$ concentrations were consistently high during the three trips (as shown by the relatively small spread of the data in Figure 8). So, we think the data are representative for the motorcycle rides on the roads. In the Taipei metropolitan area, the number of motorcycles over the past decade has increased steadily despite a more extensive MRT network. According to the data from Taiwan Ministry of Traffic and Communication, as of 31 May 2019, there are more than 3 million motorcycles in the metropolitan Taipei area (Taipei City and New Taipei City) [64]. More workers employed by the emerging service industry, for example, Black Cat Delivery Services, Food Panda and UberEATS, use motorcycles to deliver mails, packages, or food and thus may be at higher risk for PM-associated health effects since they can be riding long mileage [65]. For an UberEATS delivery person who spends 8 h a day, 5 days a week on a motorcycle, the person may inhale the cumulative exposure to $PM_{2.5}$ a year would be more than 46 mg a year, assuming a minute ventilation of 5 L/min [55]. It is difficult to measure accurately the $PM_{2.5}$ exposure during the moving moment. Many factors may affect the exposure intensity, including the wind generated during the ride and the density of the vehicles on the road. A reasonable assumption would be that when the motorcycles are moving, the riders are probably exposure to lower $PM_{2.5}$ than when the motorcycles are idle at the intersections. So, the estimated cumulative exposure could be less than what is calculated here using the concentration measured when the motorcycle is idle, but the exposure remains high.

A previous study from 2008 by Tsai et al. that compared $PM_{2.5}$ exposure in different modes of transportation showed motorcyclists had the highest exposure (67.5 $\mu g/m^3$) followed by bus riders (38.5 $\mu g/m^3$), MRT riders (35 $\mu g/m^3$) and car drivers (22.1 $\mu g/m^3$) [29].

The order of the exposure dose was like ours, except for the MRT riders. The study did not report $PM_{2.5}$ concentrations for underground and elevated MRT separately. It is possible that the $PM_{2.5}$ concentration for MRT riders reported in that study included those on the elevated sections of the tracks. The major sources for $PM_{2.5}$ for different modes of transportation include automobiles (walking, riding motorcycle, bus, car and elevated MRT) and MRT (underground MRT). The cross-border sources tend to be insignificant in the summertime in Taiwan. Air-conditioning in the bus and the car decreases the $PM_{2.5}$ concentration. The observation that $PM_{2.5}$ in the underground MRT is high in the train and on the platform and lower in the elevated MRT suggests that inadequate ventilation plays an important role. Also note that most $PM_{2.5}$ measurements during commuting were higher than the ambient concentrations (Figures 3–5). The discrepancy, in addition to the overestimation by the personal $PM_{2.5}$ detector, could be due to vertical gradient between the ground level $PM_{2.5}$ and the ambient $PM_{2.5}$ measured by the monitoring stations at a height of 10–15 m [30,49,50]. Overall, despite the expansion of the MRT network in the Taipei metropolitan area over the past decades, it seems that commuting with motorcycle and MRT can still be exposed to high concentrations of $PM_{2.5}$. The increasing numbers of motorcycles and MRT ridership raise the concern that the number of people who have high exposure is larger.

One limitation of this study is related to the hand-held detector that used an optical method to detect $PM_{2.5}$. These sensor measurements may be influenced by co-responsive pollutants, environmental conditions (e.g., humidity) and sensor component production variations [66]. High relative humidity (>80%) may result in overestimation of $PM_{2.5}$ concentration [67]. The sensor used in this study was tested in the laboratory and the field [49,50]. The results showed the climate condition had minimal effect on the precision of the sensor. We measured $PM_{2.5}$ concentrations at low wind condition. The sensor was shown to overestimate $PM_{2.5}$ concentrations. Taking the average of the three linear equations from the field test for the three detectors like ours, one may derive the following linear equation to make correction: [Adjusted $PM_{2.5}$] = 0.678 × [measured $PM_{2.5}$] + 3.298 [49]. Based on this equation, the measured $PM_{2.5}$ concentrations will need to be adjusted downward by about 30% [49]. The precision, however, was very good with low intra-modal variability (~7%) Therefore, although our detector could overestimate the absolute values of $PM_{2.5}$, the relative changes and the direction of the changes would not be affected. Our detector has also not been used in previous transportation-related studies, but other low cost light scattering sensors have been evaluated in the field for long-term monitoring up to 320 days, such as Plantower PMS 1003, PMS 5003 [68,69]. In general, the results showed good correlation with reference monitors, but there could be long-term drift in the sensor. Our study was a relatively short term one. So, the effect of drift should be minimum.

In summary, our study found subgroups of residents in the Taipei metropolitan area who may be exposed to high concentrations of $PM_{2.5}$. The exposure for the motorcyclists remained high compared to a previous study in 2008, despite the expansion of the MRT network since then. The exposure for the motorcyclists is also higher compare to the MRT riders when the train on the elevated outdoor tracks. The increase in the MRT ridership indicates more people are potentially exposed to high $PM_{2.5}$ concentrations during commuting. The very high concentration of $PM_{2.5}$ in the underground food court puts the full-time workers in high risk for PM-induced health effects. There have been no health studies on these subpopulations. Epidemiological and filed studies to assess the $PM_{2.5}$-associated health risks in these subpopulations need to be conducted in the future.

**Author Contributions:** Conceptualization, Y.-C.T.H.; methodology, C.-Y.W., B.-S.L. and Y.-H.W.; investigation, C.-Y.W., B.-S.L., Y.-C.T.H. and Y.-H.W.; resources, C.-Y.W. and Y.-C.T.H.; data curation, B.-S.L. and Y.-H.W.; writing-original draft preparation, C.-Y.W., B.-S.L. and Y.-C.T.H.; writing-review and editing, Y.-C.T.H.; supervision, Y.-C.T.H. All authors have read and agreed to the published version of the manuscript.

**Funding:** This research received no external funding.

**Institutional Review Board Statement:** Not applicable.

**Informed Consent Statement:** Not applicable.

**Data Availability Statement:** The raw data are available upon request.

**Acknowledgments:** The authors would like to thank all volunteers who participated in the study and Cardinal Tien Hospital for the logistic and funding support.

**Conflicts of Interest:** The authors declare no conflict of interest.

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
