# Peer review of "Identification of High Personal PM2.5 Exposure during Real Time Commuting in the Taipei Metropolitan Area"

_atmosphere, doi:10.3390/atmos12030396_

Round 1
Reviewer 1 Report
In the revision manuscript of Wang et al. 2021, I found that the authors have attempted to address my concerns by providing more information related to the measurement methods. However, I still feel unconvinced on a number of issues, especially the reliability of the measurement method and the representativeness of the sample collected. Also, there is a lack of hypothesis testing in this manuscript. The authors need to indicate to the reader the significance of this study as already stayed in the title that "high personal PM2.5 exposure . . . ". Following is a list of additional concerns.
Any corrections have been made to deal with the “overestimate issues” of the measurement method used in this study? Given the uncertainty among the measurement methods, it’s high uncertainty to compare the results.
Line 106-112: I don’t understand how the authors can use the air-conditioned room to validate the instruments? It’s not a standard method for calibration.
Line 111-112: “low wind condition” also applies for riding motorcycle? For this type of transportation, the sampling distribution should be very large and discrete (sampling only when motorcycle stop).
There are very few volunteers who participated in this study (5 participations) thus the representative is put into considerations.
Fig. 2: The “overestimate issues” could contribute to the discrepancy. I would expect correction factors or something to correct the data of this study.
The number of samples collected and the number of volunteers participating is relatively small to make an accurate judgment and compare among the transportation types. For example, the form of walking is greatly influenced by the route (passing through the market area or along with the shopping centers). Therefore, the authors need to clarify how can they control these factors before making the comparison.
Lines 175-178: it’s not the only reason.
Fig.4: The concentrations reported here represent one sample or average of all data collected? Similar questions for Fig. 5, 6, 7.
Line 315-317: The estimation of exposure assessment here is limited and should be interpreted more specifically and compared with other types of traffic.
Line 327: Why transport from China is insignificant during this time?
Line 345-346: Please give the evidence to support these statements.
Given that Taipei is one of the "very clean" city worldwide, thus the title makes me feel a bit curious. The authors may need to consider modifying the title.
Author Response
Comments and Suggestions for Authors
In the revision manuscript of Wang et al. 2021, I found that the authors have attempted to address my concerns by providing more information related to the measurement methods. However, I still feel unconvinced on a number of issues, especially the reliability of the measurement method and the representativeness of the sample collected. Also, there is a lack of hypothesis testing in this manuscript. The authors need to indicate to the reader the significance of this study as already stayed in the title that "high personal PM2.5 exposure . . . ". Following is a list of additional concerns.
Thanks for the comments. We have added a hypothesis in the manuscript “MRT riders in the Taipei metropolitan area were exposed to high PM2.5 during commuting”. The significance of the study was made more explicit. “The results can help identify high risk subpopulations that can be the focus of future studies on PM2.5- associated health effects”.
The questions surrounding the measurement method is addressed in comments 1 and 2 below.
1: Any corrections have been made to deal with the “overestimate issues” of the measurement method used in this study? Given the uncertainty among the measurement methods, it’s high uncertainty to compare the results.
Reply: Thank you for your comments.
Taking the average of the three linear equations from the field test for the three detectors like ours, one may derive the following linear equation to make correction: [Adjusted PM2.5] = 0.678x[measured PM2.5] + 3.298 (ref 49 in revised manuscript). Based on this equation, the measured PM2.5 concentrations will need to be adjusted downward by about 30%. The precision, however, was very good with excellent correlation of determination (average R2 = 0.915). This is added to the limitation part of the discussion (Line 337-341).
2: Line 106-112: I don’t understand how the authors can use the air-conditioned room to validate the instruments? It’s not a standard method for calibration.
Reply: Thank you for your comment. The field test showed that this detector had low intro-model variability (~7%) (ref 49 in revised manuscript). Since we don’t have access to the FEM GRIMM or TOEM, we could not do sensor calibration. We simply left the detector in an air-conditioned room to ensure that the readings were consistent before it was used. We did not mean to use it as the calibration. We have rewritten the sentences. Line 114-118
3: Line 111-112: “low wind condition” also applies for riding motorcycle? For this type of transportation, the sampling distribution should be very large and discrete (sampling only when motorcycle stop).
Reply: Thank you for your comment. For safety reason, the PM2.5 measurements were only obtained when the motorcycles were idle. This was mentioned in Line 152-153.
4: There are very few volunteers who participated in this study (5 participations) thus the representative is put into considerations.
Reply: Thank you for your comment. We have purposely kept the number of volunteers small so the measurements had less interobserver variability.
5: Fig. 2: The “overestimate issues” could contribute to the discrepancy. I would expect correction factors or something to correct the data of this study.
Reply: Thank you for your comment. This has been addressed in Comment 1.
6: The number of samples collected and the number of volunteers participating is relatively small to make an accurate judgment and compare among the transportation types. For example, the form of walking is greatly influenced by the route (passing through the market area or along with the shopping centers). Therefore, the authors need to clarify how can they control these factors before making the comparison.
Reply: As mentioned in comment 4, the exposure during each trip was the focus of our study, so there was no need to recruit many volunteers to take the measurements. In fact, the number of volunteers was kept small to decrease the interobserver variability of the measurements. The volunteers chose their usual paths when they commuted between the work place and home on foot. We did not designate the path, so the exposure scenarios were closer to the real-world experience.
7: Lines 175-178: it’s not the only reason.
Reply: We have revised the sentence. “The concentration increased at the intersection in part because of the increased automobile traffic that generated more exhaust and road dust”. Line 193-194
9: Fig.4: The concentrations reported here represent one sample or average of all data collected? Similar questions for Fig. 5, 6, 7.
Reply: Each data point represents one measurement (an average of three readings).
10: Line 315-317: The estimation of exposure assessment here is limited and should be interpreted more specifically and compared with other types of traffic.
Reply: We have revised the sentences. “The exposure for the motorcyclists remained high compared to a previous study in 2008, despite the expansion of the MRT network since then. The exposure for the motorcyclists is also higher compare to MRT riders when the train on the elevated outdoor tracks”. Line 349-352
11: Line 327: Why transport from China is insignificant during this time?
Reply: It is due to the southwest wind that prevents the smog from Mainland China to travel southward during that time.
12: Line 345-346: Please give the evidence to support these statements.
Reply: We are not sure what the reviewer meant. Line 345-346 are part of the figure legends.
13: Given that Taipei is one of the "very clean" city worldwide, thus the title makes me feel a bit curious. The authors may need to consider modifying the title.
Reply: The air quality in Taipei is affected by the cross-border sources and the local automobile traffic. Although the air quality in Taipei is better than many cities in the surrounding regions, the PM2.5 concentrations are still mostly above the WHO standard (10 ug/m3 annual mean) during the year. A reference is provided here that compared the air quality between Taipei Taiwan and Jakarta, Indonesia (file:///C:/Users/Yuh-Chin%20Huang/Downloads/ijerph-16-04924.pdf).
We have changed the title. “Identification of high personal PM2.5 exposure during real time commuting in the Taipei metropolitan area”.

Reviewer 2 Report
I find this article very interesting and with practical implication.
I have two main concerns: how many volunteers participated in experiment (I couldn't find it in the text) and if I understood correctly it was only one detector used?
Second, I think 3 bike rids is not enough for making assumptions. I know that authors use as n number of PM measurements but in my opinion comparing whit other activities is to much. Maybe elaborate on this in discussion?
Author Response
Comments and Suggestions for Authors
I find this article very interesting and with practical implication.
Thank you for the comments.
I have two main concerns: how many volunteers participated in experiment (I couldn't find it in the text) and if I understood correctly it was only one detector used?
Reply: There were five volunteers sharing one detector. We have now mentioned this in the methods. Line 120-122.
Second, I think 3 bike rids is not enough for making assumptions. I know that authors use as n number of PM measurements but in my opinion comparing whit other activities is to much. Maybe elaborate on this in discussion?
Reply: Thank you for your comment. We understand the concern. However, there were 21 measurements performed at different stops during the trips. Due to the safety concern, the motorcyclist measured PM2.5 only when the motorcycles were idle (e.g. waiting for red light). All routes were on the thoroughfares in the Taipei metropolitan area. The PM 2.5 concentrations were consistently high during the three trips (as shown by the relatively small spread of the data in Figure 8). So we think the data are representative for the motorcycle rides on the roads. This has been added in Line 289-293

Reviewer 3 Report
This study highlights the personal PM2.5 exposure in Taipei Metropolitan Area, which has useful information for considering health-related problems. Overall, the scientific approach is sound. However, few points should be addressed.
1) The motorcyclist measurement was carried out only in idle conditions for safety reasons. Would it make higher PM2.5 values from the own motorcycle emission than the actual moving period?
2) Connected to the 1) question, can authors include any information on the stopping vs. moving moments? From the health-assessment of the motorcyclist, it is essential to consider these periods.
3) The consideration of minute ventilation is useful for health assessment; however, the value may change according to the activities. A rationale to use 5 L/min. as the minute ventilation for the MRT and motorcycle riders is necessary.
Author Response
Comments and Suggestions for Authors
This study highlights the personal PM2.5 exposure in Taipei Metropolitan Area, which has useful information for considering health-related problems. Overall, the scientific approach is sound. However, few points should be addressed.
1) The motorcyclist measurement was carried out only in idle conditions for safety reasons. Would it make higher PM2.5 values from the own motorcycle emission than the actual moving period?
Reply: Thank you for your comment. That is correct. The sources of PM2.5 can come from the emission from other vehicles as well as their own motorcycle. The sentence is revised. Line 277-278.
2) Connected to the 1) question, can authors include any information on the stopping vs. moving moments? From the health-assessment of the motorcyclist, it is essential to consider these periods.
It is difficult to measure accurately the PM2.5 exposure during the moving moment. Many factors may affect the exposure intensity, including the wind generated during the ride and the density of the vehicles on the road. A reasonable assumption would be that when the motorcycles are moving, the riders are probably exposure to lower PM2.5 than when the motorcycles are idle at the intersections. So the estimated cumulative exposure could be less than what is calculated here using the concentration measured when the motorcycle is idle, but the exposure remains high. This information has been added. Line 304-310
3) The consideration of minute ventilation is useful for health assessment; however, the value may change according to the activities. A rationale to use 5 L/min. as the minute ventilation for the MRT and motorcycle riders is necessary.
Reply: Thank you for your comment. A resting minute ventilation is used for the calculation because when the PM2.5 concentrations were measured, the MRT riders were stationary. The minute ventilation can increase when they walk around increasing the inhalation dose. So the calculation represents a conservative estimate. We have added this to the discussion. Line 251-254

Reviewer 4 Report
The paper presents an interesting comparison of PM2.5 concentrations during commuting in different modes of transportation in the Taipei metropolitan area (Taiwan). This allowed the authors to identify which mode of transport was associated with the greatest exposure to the adverse effects of PM2.5. The conducted research provides interesting insights, the literature review and discussion of the results are extensive. The paper is well written, with interesting results, but requires some additional details before final publication.
Lines 1-2: Manuscript title should be rethought.The work is mainly devoted to PM2.5 concentrations in real time during commuting in different modes of transportation. The exposure assessment is very simplified.Moreover, exposure was calculated only for two groups (MRT rider and ubereats delivery person).What about the rest (pedestrians, motorcyclists)?
Lines 65-69: Are these findings from the conducted research? If so, should be moved to the results discussion, if not then please add references.
Lines 107, 130-133, 169-171, 208, 252: The authors report how many trips have been made, but it is more crucial to know how many measurements have been taken? How long did each measurement take? Maybe it is worth presenting it in the form of a table? The authors should explain the measurement methodology in more detail.
Line 170-174: How long did these measurements take? Please explain to the reader why this is a comparison of the two measurement methods.
Line 268: Please explain the reason for this assumption.Please add appropriate references. I suggest that the exposure assessment description should be included in the methodology.Especially that this term appears in the title of the paper.
Novelty: The topic of high PM concentrations in metropolitan areas has been very popular for many years. There are many works devoted to similar topics. Authors should explain what is new and different in their manuscript.
Minor:
Figure 3: Should be: Measurement points
Author Response
Comments and Suggestions for Authors
The paper presents an interesting comparison of PM2.5 concentrations during commuting in different modes of transportation in the Taipei metropolitan area (Taiwan). This allowed the authors to identify which mode of transport was associated with the greatest exposure to the adverse effects of PM2.5. The conducted research provides interesting insights, the literature review and discussion of the results are extensive. The paper is well written, with interesting results, but requires some additional details before final publication.
Thank you for the comments.
Lines 1-2: Manuscript title should be rethought. The work is mainly devoted to PM2.5 concentrations in real time during commuting in different modes of transportation. The exposure assessment is very simplified. Moreover, exposure was calculated only for two groups (MRT rider and ubereats delivery person). What about the rest (pedestrians, motorcyclists)?
Reply: we change the title as: “Identification of high personal PM2.5 exposure during real time commuting in the Taipei metropolitan area”. We only calculated the cumulative exposure for the MRT riders and the UberEAT persons who use motorcycles because these two groups have higher PM2.5 exposure compared to other modes of transportation. The estimate for the pedestrians can be made, but a different minute ventilation would have to be assumed depending on how fast they walk. The cumulative exposure for the car drivers and the bus riders could use the resting minute ventilation. Interested readers can use our PM2.5 results to deduce the cumulative exposure using their estimate of time in each mode of transportation and the minute ventilation.
Lines 65-69: Are these findings from the conducted research? If so, should be moved to the results discussion, if not then please add references.
Reply: References are now given for these statements.
Lines 107, 130-133, 169-171, 208, 252: The authors report how many trips have been made, but it is more crucial to know how many measurements have been taken? How long did each measurement take? Maybe it is worth presenting it in the form of a table? The authors should explain the measurement methodology in more detail.
Reply: In the result section, we have indicated the number of measurements for each mode of transportation under the respective subheadings. We think this is adequate in presenting the number of measurements. Each measurement took 1 min.
Line 170-174: How long did these measurements take? Please explain to the reader why this is a comparison of the two measurement methods.
Reply: Each measurement took 1 min. The discrepancy could also be due to the different methods used to measure PM2.5. The personal PM2.5 levels were measured by the light scattering method while the ambient PM2.5 levels were measured at each monitoring station by the TOEM method by Taiwan EPA. This was added to the results. Line 185-188
Line 268: Please explain the reason for this assumption. Please add appropriate references. I suggest that the exposure assessment description should be included in the methodology. Especially that this term appears in the title of the paper.
Reply: Line 268 is a statement that “Motorcycle is a common mode of transportation in many tropical and subtropical Asian countries, including Taiwan”. This is a fact not an assumption. We have listed references in the text. The exposure assessment is calculated based on several assumptions. We think it is more appropriate that the exposure assessment be in the discussion. Interested readers can calculate their own exposure levels using our measured data for their specific commuting scenarios.
Novelty: The topic of high PM concentrations in metropolitan areas has been very popular for many years. There are many works devoted to similar topics. Authors should explain what is new and different in their manuscript.
Reply: The reviewer is correct that there have been studies on personal PM2.5 exposure in cities that have different transportation infrastructures. These cities are also located in different geographic regions with different climates. Their citizens have different lifestyles and cultures and tend to use modes of transport that are most convenient and economical. Therefore, it is essential to characterize the exposure to air pollutants in a specific urban environment so the health risk can be more accurately assessed. We have added this to the introduction. Line 84-89
Minor:
Figure 3: Should be: Measurement points
Revised. Line 199.

Reviewer 5 Report
Abstract:
-I'm not sure about what the authors want to express with "There have been increase in", maybe... there has been an increase?
Introduction:
The introduction provides enough information and references, achieving a correct reader comprehension about the actual problem treated in the present research.
-Paragraph 36-42, the authors used so many times the noun/verb increase, please use synonyms of it like "rise, growth, intensify, etc."
-Line 41, please unify the references [12-17].
Materials and Methods:
The materials and methods section provides the information required to understand how the research has been carried out, the objects of study, and the methodologies followed.
Results:
I consider that the results obtained by the authors supported the hypothesis suggested. The authors have successfully evaluated the exposure of Taipei residents to PM2.5 in different transportations modes with interesting results that can be used to change the ventilation modes of some structures with the aim of decrease the health consequences for residents. In addition, the results have been correctly and ordered presented enhancing the reading comprehension.
References: they are not in the Atmosphere journal format, please correct them.
- Journal Articles:
- Author 1, A.B.; Author 2, C.D. Title of the article. Abbreviated Journal NameYear, Volume, page range.
- Books and Book Chapters:
- Author 1, A.; Author 2, B. Book Title, 3rd ed.; Publisher: Publisher Location, Country, Year; pp. 154–196.
3. Author 1, A.; Author 2, B. Title of the chapter. In Book Title, 2nd ed.; Editor 1, A., Editor 2, B., Eds.; Publisher: Publisher”
Author Response
#Reviewer5
Comments and Suggestions for Authors
Abstract:
-I'm not sure about what the authors want to express with "There have been increase in", maybe... there has been an increase?
Reply: Revised.
Introduction:
The introduction provides enough information and references, achieving a correct reader comprehension about the actual problem treated in the present research.
Thank you for the comments.
-Paragraph 36-42, the authors used so many times the noun/verb increase, please use synonyms of it like "rise, growth, intensify, etc."
Reply: Revised.
-Line 41, please unify the references [12-17].
Reply: Revised.
Materials and Methods:
The materials and methods section provides the information required to understand how the research has been carried out, the objects of study, and the methodologies followed.
Thank you for the comments.
Results:
I consider that the results obtained by the authors supported the hypothesis suggested. The authors have successfully evaluated the exposure of Taipei residents to PM2.5 in different transportations modes with interesting results that can be used to change the ventilation modes of some structures with the aim of decrease the health consequences for residents. In addition, the results have been correctly and ordered presented enhancing the reading comprehension.
Thank you for the comments.
References: they are not in the Atmosphere journal format, please correct them.
Journal Articles:
Author 1, A.B.; Author 2, C.D. Title of the article. Abbreviated Journal NameYear, Volume, page range.
Books and Book Chapters:
Author 1, A.; Author 2, B. Book Title, 3rd ed.; Publisher: Publisher Location, Country, Year; pp. 154–196.
- Author 1, A.; Author 2, B. Title of the chapter. In Book Title, 2nd ed.; Editor 1, A., Editor 2, B., Eds.; Publisher: Publisher”
Reply: Thank you for your comment and we corrected the format.

Round 2
Reviewer 1 Report
I feel that the authors have been revised the manuscript intensively and carefully addressed most of my major concerns in this revision. This revision seems to be better than the previous one.
However, I still curious about the comparison of PM2.5 concentrations between this study and others. Acknowledge the high uncertainty (more than 30%, Line 381), I'm not clear whether the data in this study have been adjusted or not? If not, all of the comparison discussions (this study vs other studies, e.g., Lines 281-285) need to be removed/modified since the absolute PM2.5 values are not corrected.
In addition, intro-model uncertainty only guarantees accuracy while intra-model complements the precision of the measurement of PM2.5. I wonder both are shown 7% different? (Line 120 vs Line 126). Please specify the intra-model comparison (could be from a previous study) here to convince the readers of the method used in this study.
Author Response
I feel that the authors have been revised the manuscript intensively and carefully addressed most of my major concerns in this revision. This revision seems to be better than the previous one.
Reply: Thank you for your comment.
However, I still curious about the comparison of PM2.5 concentrations between this study and others. Acknowledge the high uncertainty (more than 30%, Line 381), I'm not clear whether the data in this study have been adjusted or not? If not, all of the comparison discussions (this study vs other studies, e.g., Lines 281-285) need to be removed/modified since the absolute PM2.5 values are not corrected.
Reply: We appreciate your additional comments. We have done the following:
- Add the linear regression equation calculated from ref 49 to the Methods (Line111-116) so the readers can make their own adjustment.
- Line 249-251. We changed the sentence to “With the 30% downward adjustment for the PM2.5 concentration measured by our portable detector, these results were slightly lower than those from a previous study that showed an average PM2.5 concentrations of 75.4 µg/m3 in the winter and 56.2 µg/m3 in the summer on the platforms of 10 most populous underground MRT stations in Taipei [54]”. Note that the average concentration in our study after 30% adjustment would be 54.3 µg/m3. The average concentration reported in that study was 65.8 µg/m3.
In addition, intro-model uncertainty only guarantees accuracy while intra-model complements the precision of the measurement of PM2.5. I wonder both are shown 7% different? (Line 120 vs Line 126). Please specify the intra-model comparison (could be from a previous study) here to convince the readers of the method used in this study.
Reply: It should be the intra-model variability in the text. We have corrected the typo.
Reviewer 4 Report
You should unify the "PM2.5" notation. "2.5" is subscript in some places and not in others. Similarly, units - please check the notation of cubic meters.
Line 22: Please note, that PM concentration is not the same as exposure. The exposure depends on the time of exposure, PM concentration, parameters of breathing and body efficiency. Instead of "exposure" you should use the word "concentration".
Line 273: What does „personal PM2.5 concentrations” mean?
Line 287: Please add appropriate references on the basis of which you assumed the minute ventilation and calculated the exposure.There are different methodologies for exposure assessment.You should refer to the one on which your calculations are based.
Author Response
You should unify the "PM2.5" notation. "2.5" is subscript in some places and not in others. Similarly, units - please check the notation of cubic meters.
Reply: Thank you for the comments. We have corrected them.
Line 22: Please note, that PM concentration is not the same as exposure. The exposure depends on the time of exposure, PM concentration, parameters of breathing and body efficiency. Instead of "exposure" you should use the word "concentration".
Reply: Thank you for the comments. We have corrected them where appropriate.
Line 273: What does „personal PM2.5 concentrations” mean?
Reply: Thank you for your comments. We have removed “personal”.
Line 287: Please add appropriate references on the basis of which you assumed the minute ventilation and calculated the exposure.There are different methodologies for exposure assessment.You should refer to the one on which your calculations are based.
Reply: Thank you for your comments. We agree. What we calculated was the estimated amount of inhaled PM2.5. So we have now used “inhaled amount” rather than “exposure”. The calculation method is based on US EPA ExpoBox (https://www.epa.gov/expobox/exposure-assessment-tools-routes-inhalation). This is added as a reference. Line 253-255, Line 305-307.
This manuscript is a resubmission of an earlier submission. The following is a list of the peer review reports and author responses from that submission.
Round 1
Reviewer 1 Report
In Figure 2, some points deviate quite significantly from the regression line. It would be necessary to give an explanation about this.
Author Response
We have added an explanation to Figure 2. The poor association was in part due to the difference in the measurement of PM2.5. The PM2.5 reported by the monitoring station was measured at a height of 10-15 meters while the PM2.5 measured by the hand-held detector was the ground level. It is known that the vertical gradient of PM2.5 concentration can be as much as 10-20 fold. A reference is added.
Reviewer 2 Report
General remarks
The Introduction should explain why the question tested by the experiments described in the paper is an interesting or important question, describe the approach used in sufficient detail that a reader who is not familiar with the technique will understand what was done and why, and very briefly mention the conclusion of the paper.
I believe that adequate background is missing in this section. According to the authors, the goal is to identify high-risk subpopulations that will be the focus of future studies on PM2.5 associated health effects. Because of that, I suggest that the impact of air pollution exposure on health is briefly addressed in the Introduction (see for example Hoek G, Krishnan RM, Beelen R, Peters A, Ostro B, Brunekreef B, Kaufman JD (2013) Long-term air pollution exposure and cardiorespiratory mortality: a review. Environ Health 12:1.; Tseng E, Ho W-C, Lin M-H, Cheng T-J, Chen P-C, Lin H-H (2015) Chronic exposure to particulate matter and risk of cardiovascular mortality: cohort study from Taiwan. BMC Public Health 15:1; Zuurbier M, Hoek G, Oldenwening M, Meliefste K, van den Hazel P, Brunekreef B (2011) Respiratory effects of commuters’ exposure to air pollution in traffic. Epidemiology 22:219–227; Chen H, Goldberg MS, Villeneuve PJ. A systematic review of the relation between long-term exposure to ambient air pollution and chronic diseases. Rev Environ Health. 2008 Oct-Dec;23(4):243-97.)
Also, similar research should be mentioned in the Introduction, not in the Discussion section (references 10–32).
According to the Instructions for authors, only the Results section should be divided into subsections. Since all subsections in the section Materials and methods are very brief, I believe that subsections are not needed in this section.
In the Discussion section limitations of the work should be highlighted. This section should explore the significance of the results of the work, not repeat them. Also, future research directions should be mentioned, and similar research that is addressed in this section should be moved to the Introduction section.
Since the Conclusion section is very brief, and some conclusions are already addressed in the Discussion section (for example lines 258–259, 276–278), I think that this section should form a part of a Discussion.
According to the Instructions for authors, websites should be listed as references - “Websites: 9. Title of Site. Available online: URL (accessed on Day Month Year).” (see lines 54, 91, 96, 177, 190, 206, 217, 229, 294).
For clarity's sake, keep the titles of Figures and Tables short and concise - avoid placing long descriptions in their titles, describe them in the main text instead.
Specific remarks
Line 52: acronyms like MRT should be explained when they first appear.
Line 67: unnecessary.
Lines 75–76: Figure 1 is unnecessary – it does not benefit the location description as it does not provide sufficient information on the area in question.
Line 77: unnecessary.
Lines 138–139: format the equation according to the Instructions for authors.
Lines 142–143: place the description in the main text.
Lines 159–161: place the description in the main text (for example in Line 153: “Table 1 shows an example of a short walk from home to the workplace (hospital) on a sunny day with a reported temperature of 26°C and relative humidity of 69%.)
Lines 162–164, 175–179, 189–191, 205–207, 215–218: place the description in the main text.
Line 174: the table presented in Figure 3 should be an independent entity (i.e., Table 3. PM2.5 concentrations during a walk across a business district).
Figures 4–7: indicate measuring units for “Time”.
Lines 192, 207, 218, 230: place the legend (MRT ride, bus ride, walking) in the Figure, not in the Figure title.
Lines 237–246: place the description in the main text.
Line 301–302: “Studies to assess the PM2.5-associated health risks in these subpopulations are urgently needed.” – explain why these studies are urgently needed and explain the importance of study results to the possible audience.
Author Response
The Introduction should explain why the question tested by the experiments described in the paper is an interesting or important question, describe the approach used in sufficient detail that a reader who is not familiar with the technique will understand what was done and why, and very briefly mention the conclusion of the paper.
Revised
I believe that adequate background is missing in this section. According to the authors, the goal is to identify high-risk subpopulations that will be the focus of future studies on PM2.5 associated health effects. Because of that, I suggest that the impact of air pollution exposure on health is briefly addressed in the Introduction (see for example Hoek G, Krishnan RM, Beelen R, Peters A, Ostro B, Brunekreef B, Kaufman JD (2013) Long-term air pollution exposure and cardiorespiratory mortality: a review. Environ Health 12:1.; Tseng E, Ho W-C, Lin M-H, Cheng T-J, Chen P-C, Lin H-H (2015) Chronic exposure to particulate matter and risk of cardiovascular mortality: cohort study from Taiwan. BMC Public Health 15:1; Zuurbier M, Hoek G, Oldenwening M, Meliefste K, van den Hazel P, Brunekreef B (2011) Respiratory effects of commuters’ exposure to air pollution in traffic. Epidemiology 22:219–227; Chen H, Goldberg MS, Villeneuve PJ. A systematic review of the relation between long-term exposure to ambient air pollution and chronic diseases. Rev Environ Health. 2008 Oct-Dec;23(4):243-97.)
Revised
Also, similar research should be mentioned in the Introduction, not in the Discussion section (references 10–32).
Revised
According to the Instructions for authors, only the Results section should be divided into subsections. Since all subsections in the section Materials and methods are very brief, I believe that subsections are not needed in this section.
Revised
In the Discussion section limitations of the work should be highlighted. This section should explore the significance of the results of the work, not repeat them. Also, future research directions should be mentioned, and similar research that is addressed in this section should be moved to the Introduction section.
Revised
Since the Conclusion section is very brief, and some conclusions are already addressed in the Discussion section (for example lines 258–259, 276–278), I think that this section should form a part of a Discussion.
Revised
According to the Instructions for authors, websites should be listed as references - “Websites: 9. Title of Site. Available online: URL (accessed on Day Month Year).” (see lines 54, 91, 96, 177, 190, 206, 217, 229, 294).
Revised
For clarity's sake, keep the titles of Figures and Tables short and concise - avoid placing long descriptions in their titles, describe them in the main text instead.
Revised
Specific remarks
Line 52: acronyms like MRT should be explained when they first appear.
Revised
Line 67: unnecessary.
Removed.
Lines 75–76: Figure 1 is unnecessary – it does not benefit the location description as it does not provide sufficient information on the area in question.
We have changed Figure 1 as the suggestion of you and Reviewer 1.
Line 77: unnecessary.
Removed.
Lines 138–139: format the equation according to the Instructions for authors.
Revised
Lines 142–143: place the description in the main text.
Revised
Lines 159–161: place the description in the main text (for example in Line 153: “Table 1 shows an example of a short walk from home to the workplace (hospital) on a sunny day with a reported temperature of 26°C and relative humidity of 69%.)
Revised
Lines 162–164, 175–179, 189–191, 205–207, 215–218: place the description in the main text.
Revised
Line 174: the table presented in Figure 3 should be an independent entity (i.e., Table 3. PM2.5 concentrations during a walk across a business district).
Revised
Figures 4–7: indicate measuring units for “Time”.
Revised
Lines 192, 207, 218, 230: place the legend (MRT ride, bus ride, walking) in the Figure, not in the Figure title.
Revised
Lines 237–246: place the description in the main text.
Revised
Line 301–302: “Studies to assess the PM2.5-associated health risks in these subpopulations are urgently needed.” – explain why these studies are urgently needed and explain the importance of study results to the possible audience.
Revised

Reviewer 3 Report
The manuscript of Wang et al. 2021 provides a 4-months personal exposure measurement of PM2.5 associated with different modes of transportation in Taipei, Taiwan. The major aims of this study are to provide an updated estimate of personal PM2.5 exposure and identify high-risk sub-populations that can be the focus of future studies on PM2.5 associated health effects. This study is an updated study from Tsai et al. (2008) with similar aims, thus comparison discussions are needed. Lack of in-deep discussions related to the driving factors behind the observation results. The discussions associated with the measurement method need to be strengthened to convince the readers about the suitability of the method used in this study. Following is a list of additional concerns.
- Abstract:
I suggest the authors present the results of PM2.5 concentrations via walking before comparing this to other types of transportation (i.e. riding motorcycle and temples).
This study focus on the personal PM2.5 exposure, however, no exposure assessment (e.g. personal average exposure amount values or similar indicators) results were reported in the abstract. Estimation results from the PM2.5 exposure could be valuable and need to be discussed.
- Introduction:
Line 47-48: Could the authors provide more comprehensive discussions of the results from previous studies to indicate the impacts of different modes of transportation on PM2.5 exposure? Which mode is assessed as the riskiest for PM2.5 exposure?
An introduction paragraph of previous methods (i.e. PM measurements) that have been used to conduct such kind of study could be helpful.
- Materials and methods:
Line 78-83. More details related to the PM2.5 detector (e.g., routine calibration procedure, data QA/QC, data time resolution) need to provide here to support the measurement method. How many samples have been collected from this study? Besides, add references to indicate the suitability of the measurement method used in this study.
Line 83. Clarify the sentence: “Only one detector was used for the entire study”. Any meteorological factors can influence the precision of the measurement method, particularly wind speed?
Line 84-86. The authors need to provide data/references to support these statements.
Line 86-96. How many participants join in this study? Also, please provide some general biological characteristics of the participants. I would suggest adding the sampling locations of EPA stations used in this study into Fig. 1.
Line 97-120. How many participants for each mode of transportation? Any daily activity diary was conducted to collect information about the study participant’s activity pattern?
Fig. 1. I would suggest providing a zoom-in map for MRT lines along with the general map.
- Results
Fig. 2. Add the linear equation, p-value and R2 onto this figure. Fig. 2 shows a very poor correlation between street level and monitoring station PM2.5, what are the reasons to explain this discrepancy?
Line 148-150. Provide a figure to support this statement.
Fig. 3, 4 and 5. Given the huge bias between EPA station and personal PM2.5 detector, the authors need to explain the driving factors of such discrepancy.
Line 210: p = 0.0025 vs walking. Higher or lower? Also, please check and clarify this throughout the manuscript.
Fig.8. The authors need to adjust the scale of this graph.
I also suggest the authors improving the clarity of the figures reported in this study.
Given that the personal inhalation rates should be different among the modes of transportation leading to a discrepancy in health impacts. Therefore, the authors need to consider to include the personal average exposure amount assessment in this study.
- Discussion
Reasons associated with the elevated PM2.5 concentration underground MRT need to be discussed and compare to other studies worldwide. Are there other studies that used the same method as this study to measure PM2.5?
Could the authors explain the mechanisms leading to greater PM2.5 exposure during riding than walking and bus?
Lack of discussion and/or explanation of the driving factors (sources, meteorological, etc.) associated with the difference in PM2.5 concentrations observed in Fig. 8.
Major concerns:
In this study, the authors employed the Temtop P600 Air Quality Laser Particle Detector to measure the PM2.5 concentrations. Based on the comparison to the traditional measurement method (TEOM method- EPA), the relationship between the two measurement methods is very poor R=0.524 (i.e. R2=0.27) and the intercept is 23.35 ug/m3, triggering a critical concern on the data reported throughout the study as well as the comparison discussions. Authors need to demonstrate the reliability of the method used in this study before it can be used to discuss. Given that the sensor method to measure PM2.5 is strongly influenced by meteorological conditions, especially RH, the authors need to demonstrate that the impacts of these factors are small and not affect the final measurement results.
Any direct side by side comparison between Temtop P600 and TEOM has been conducted?
Discussions associated with the driving factors behind the observation results are insufficient and need to be improved. Since this study is an updated study of Tsai et al. (2008), readers should expect more comparison discussion between these two studies, which is lacking in this version of the manuscript.
In addition, this manuscript mainly discusses the measurement of PM2.5 concentrations (ug/m-3) along with various modes of transportation rather than attempt to investigate the PM2.5 exposure amount.
Author Response
1. Abstract:
I suggest the authors present the results of PM2.5 concentrations via walking before comparing this to other types of transportation (i.e. riding motorcycle and temples).
Done.
This study focus on the personal PM2.5 exposure, however, no exposure assessment (e.g. personal average exposure amount values or similar indicators) results were reported in the abstract. Estimation results from the PM2.5 exposure could be valuable and need to be discussed.
The PM2.5 concentrations measured by the portable detector were what the subjects would be exposed to. We have added the estimated cumulative exposure dose in the Discussion.
2. Introduction:
Line 47-48: Could the authors provide more comprehensive discussions of the results from previous studies to indicate the impacts of different modes of transportation on PM2.5 exposure? Which mode is assessed as the riskiest for PM2.5 exposure?
The traffic patterns, traffic volumes and driving conditions may explain part of the variabilities. For example, driving a car was exposed to higher PM, but if the windows were closed with air conditioning on, the PM concentration inside the car decreased. When biking routes shared the road with car lanes, especially when the traffic volumes were high, the bikers got higher exposure. This has been added to the paragraph.
An introduction paragraph of previous methods (i.e. PM measurements) that have been used to conduct such kind of study could be helpful.
Done. Different methods, mostly light scattering and gravimetric techniques were used in previous studies.
3. Materials and methods:
Line 78-83. More details related to the PM2.5 detector (e.g., routine calibration procedure, data QA/QC, data time resolution) need to provide here to support the measurement method. How many samples have been collected from this study? Besides, add references to indicate the suitability of the measurement method used in this study.
Done. The detector was calibrated in the office which has continuous PM2.5 measurements by a gravimetric sensor before the trip. We used outliers and negative values to do the basic QA/QC.
Line 83. Clarify the sentence: “Only one detector was used for the entire study”. Any meteorological factors can influence the precision of the measurement method, particularly wind speed?
We used the same detector for the entire study to minimize the variability among different units, although the variability is small (~7%). The climate condition had minimal effect on the sensors’ precision, up to 65% relative humidity. We measured PM2.5 concentrations at low wind speed. We have included this information in the limitation.
Line 84-86. The authors need to provide data/references to support these statements.
We have added a reference.
Line 86-96. How many participants join in this study? Also, please provide some general biological characteristics of the participants. I would suggest adding the sampling locations of EPA stations used in this study into Fig. 1.
Line 97-120. How many participants for each mode of transportation? Any daily activity diary was conducted to collect information about the study participant’s activity pattern?
There were 2-3 participants for each mode of transportation. Their tasks were to measure PM2.5 and record environmental conditions during the trip. The participants did not record their daily activity or keep a diary.
Fig. 1. I would suggest providing a zoom-in map for MRT lines along with the general map.
We changed Figure 1 as the suggestion of you and Reviewer 2.
4. Results
Fig. 2. Add the linear equation, p-value and R2 onto this figure. Fig. 2 shows a very poor correlation between street level and monitoring station PM2.5, what are the reasons to explain this discrepancy?
The poor correlation may in part reflect the differences between street level PM2.5 and ambient PM2.5 measured by the monitoring stations that are 10-15 meter high. A reference is added.
Line 148-150. Provide a figure to support this statement.
Instead of adding a figure, we chose to delete the sentence as it seems irrelevant to the current study.
Fig. 3, 4 and 5. Given the huge bias between EPA station and personal PM2.5 detector, the authors need to explain the driving factors of such discrepancy.
The difference, in addition to the overestimation by the personal PM2.5 detector, could be due to differences between ground level PM2.5 and ambient PM2.5 measured by the monitoring stations.
Line 210: p = 0.0025 vs walking. Higher or lower? Also, please check and clarify this throughout the manuscript.
Done
Fig.8. The authors need to adjust the scale of this graph.
I also suggest the authors improving the clarity of the figures reported in this study.
Given that the personal inhalation rates should be different among the modes of transportation leading to a discrepancy in health impacts. Therefore, the authors need to consider to include the personal average exposure amount assessment in this study.
Done
5. Discussion
Reasons associated with the elevated PM2.5 concentration underground MRT need to be discussed and compare to other studies worldwide. Are there other studies that used the same method as this study to measure PM2.5?
Done.
Could the authors explain the mechanisms leading to greater PM2.5 exposure during riding than walking and bus?
Done
Lack of discussion and/or explanation of the driving factors (sources, meteorological, etc.) associated with the difference in PM2.5 concentrations observed in Fig. 8.
Done
Major concerns:
In this study, the authors employed the Temtop P600 Air Quality Laser Particle Detector to measure the PM2.5 concentrations. Based on the comparison to the traditional measurement method (TEOM method- EPA), the relationship between the two measurement methods is very poor R=0.524 (i.e. R2=0.27) and the intercept is 23.35 ug/m3, triggering a critical concern on the data reported throughout the study as well as the comparison discussions. Authors need to demonstrate the reliability of the method used in this study before it can be used to discuss. Given that the sensor method to measure PM2.5 is strongly influenced by meteorological conditions, especially RH, the authors need to demonstrate that the impacts of these factors are small and not affect the final measurement results.
The poor correlation between the portable detector and the ambient PM2.5 was not unexpected. PM2.5 concentrations near the ground could be 10-20 times higher than those found 10 meter above the ground (Wu C-F, et al. Environ Res 133:96–102).
Any direct side by side comparison between Temtop P600 and TEOM has been conducted?
The sensor was tested in the lab and the field and compared with FEM-GRIMM. It showed excellent correlation with FEM-GRIMM. The climate condition had minimal effect on the sensors’ precision. We have added the references.
Discussions associated with the driving factors behind the observation results are insufficient and need to be improved. Since this study is an updated study of Tsai et al. (2008), readers should expect more comparison discussion between these two studies, which is lacking in this version of the manuscript.
Done.
In addition, this manuscript mainly discusses the measurement of PM2.5 concentrations (ug/m-3) along with various modes of transportation rather than attempt to investigate the PM2.5 exposure amount.
Done.

Round 2
Reviewer 2 Report
The authors did nothing to improve the quality of the presentation of their research.
Reviewer 3 Report
Although this is the second version of the manuscript, I think it no improvement compared to the first version in terms of scientific meaning and presentations. Also, the authors only respond to the reviewer with very limited information, superficial, and do not sufficiently revise the contents of the manuscript. More importantly, the authors only said they "done" for my comments and revise/remove information in the revised version but I discover that most of them were not revised and are the same as the first version. Especially, the discussion section remains the same, the authors mentioned that they already added discussions/estimations related to the cumulated expose dose in the discussion but I didn't see any information in the discussion section. Therefore, I do not satisfy this revision and can not recommend it for publication.